# On Testability of the Front-Door Model via Verma Constraints

**Rohit Bhattacharya**[1]                    **Razieh Nabi**[2]

[1]Department of Computer Science, Williams College, Williamstown, Massachusetts, USA
[2]Department of Biostatistics and Bioinformatics, Emory University, Atlanta, Georgia, USA

## Abstract

The front-door criterion can be used to identify and compute causal effects despite the existence of unmeasured confounders between a treatment and outcome. However, the key assumptions – (i) the existence of a variable (or set of variables) that fully mediates the effect of the treatment on the outcome, and (ii) which simultaneously does not suffer from similar issues of confounding as the treatment-outcome pair – are often deemed implausible. This paper explores the testability of these assumptions. We show that under mild conditions involving an auxiliary variable, the assumptions encoded in the front-door model (and simple extensions of it) may be tested via generalized equality constraints a.k.a Verma constraints. We propose two goodness-of-fit tests based on this observation, and evaluate the efficacy of our proposal on real and synthetic data. We also provide theoretical and empirical comparisons to instrumental variable approaches to handling unmeasured confounding.

## 1  INTRODUCTION

Adjustment on a set of observed covariates satisfying the backdoor condition [Pearl, 1995a] is a common strategy for estimating causal effects from observational data. However, in many practical scenarios, it may be impossible to find covariates satisfying this condition due to the presence of one or more unmeasured confounders affecting both the treatment and outcome. Two alternatives have received attention in the literature: (i) front-door adjustment [Pearl, 1995a] and (ii) instrumental variable methods [Wright, 1928, Balke and Pearl, 1993, Angrist et al., 1996]. Prior work has focused on proposing criteria to ensure reliability of effect estimates obtained from instrumental variable (IV) models e.g., via falsification of its assumptions [Pearl, 1995b, Wang et al.,

2017, Finkelstein et al., 2021], or in special cases, confirmation in over-identified models; see Kitagawa [2015] for an overview. In contrast, little attention has been given to proposing restrictions on the observed data that falsify or confirm the assumptions of the front-door model. Such criteria are important, as when front-door adjustment is possible, an analyst may prefer to use it over IV methods, which do not always yield point identification of the causal effect, or may impose extra restrictions (e.g., effect homogeneity) beyond the structural assumptions of the model. Empirical evaluations also suggest that front-door adjustment can recover reasonable estimates of causal effects in real-world settings where unmeasured confounding is to be expected [Glynn and Kashin, 2013, 2018, Bellemare et al., 2019].

While front-door adjustment offers an appealing alternative in settings where standard covariate adjustment is not possible, several authors have cast doubt on whether the assumptions encoded by the model are plausible in practice [Cox and Wermuth, 1995, Koller and Friedman, 2009, Imbens, 2020]. In this work, we aim to bridge the gap in testability of the front-door model. Our contributions can be summarized as follows: (i) We show that if a particular (well-known) generalized equality constraint a.k.a Verma constraint [Verma and Pearl, 1990, Spirtes et al., 2000] holds in the observed data distribution between an "anchor" variable and the outcome, it is sufficient to ensure that the assumptions of the front-door model are satisfied; (ii) We propose ways of testing this constraint with finite samples. The tests rely on variationally independent pieces of a natural parameterization of the observed likelihood, and have the appealing property that they require little additional modeling than what is typically used in inverse probability weighted estimators for the front-door functional proposed by Fulcher et al. [2020] and Bhattacharya et al. [2020]. That is, models used to perform the test can be re-used for downstream causal effect estimation (if the test indicates it is ok to proceed); (iii) Finally, we provide theoretical and empirical comparisons between IV and front-door models. We show that our proposed criterion for testing the front-door

*Accepted for the 38th Conference on Uncertainty in Artificial Intelligence* (UAI 2022).

assumptions can be combined with a simple conditional independence test that enables testing the validity of the anchor variable as an instrument. That is, we show it is possible to test an intersection model where both the IV and front-door conditions are met; we hope this opens avenues for future research into combining estimates from the two adjustment strategies with certain robustness properties.

**Related work:** Works like Maathuis et al. [2009] and Malinsky and Spirtes [2017] apply causal discovery methods (for systems with and without latent confounders respectively) to identify sets of variables that might satisfy the backdoor condition with respect to various treatment-outcome pairs. Entner et al. [2013] and Shah et al. [2022] propose an ordinary independence criterion that uses an anchor variable to determine if a set of pre-treatment covariates satisfy the backdoor condition with respect to a given treatment-outcome pair (these techniques avoid running an entire causal discovery search procedure.) These works (and others regarding testing the validity of IVs cited in the introduction) are most similar to our own, except we define a criterion that uses a generalized equality constraint involving the anchor variable to determine whether a proposed set of mediators satisfy the front-door conditions. To our knowledge, the use of Verma constraints for this purpose has not been explored before. With regards to procedures for testing Verma constraints, one of the inverse weighting procedures we propose uses different pieces of the model likelihood than what is typically used in the phrasing of the constraint; the second procedure represents a stabilized version [Hernán and Robins, 2006] of the usual weights used in Verma tests. Our methods also complement work by Thams et al. [2021] who proposed a weighted resampling scheme for producing pseudo-datasets that mimic a post-intervention distribution such that applying any (potentially non-parametric) conditional independence test to the pseudo-dataset amounts to testing the Verma constraint itself. That is, the methods of weight generation we propose can be plugged into the resampling schemes of Thams et al. [2021] to produce distinct non-parametric tests; we expand on this in future sections.

## 2 PROBLEM SETUP & MOTIVATION

Consider a setting where the analyst is interested in computing the causal effect of smoking (treatment $A$) on developing coronary heart disease (outcome $Y$). A common target of interest to quantify such effects is the mean contrast in outcomes under two different (hypothetical) interventions. More formally, the *average causal effect* (ACE) can be defined as the contrast $\mathbb{E}[Y \mid \mathrm{do}(a)] - \mathbb{E}[Y \mid \mathrm{do}(a')]$, where $\mathrm{do}(\cdot)$ denotes an intervention [Pearl, 2009]. The ACE may be identified as a function of observed data given sufficient restrictions on a causal model. For example, given a set of covariates $C$, the ignorability assumption $Y(\mathrm{do}(a)) \perp\!\!\!\perp A|C$, along with positivity of the distribution $p(A|C)$ and consis-

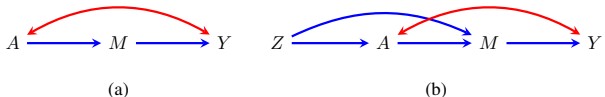

Figure 1: (a) The front-door model; (b) The front-door model with an anchor variable $Z$.

tency, yields identification of the ACE via the *adjustment formula*: $\mathbb{E}[\mathbb{E}[Y|A = a, C] - \mathbb{E}[Y|A = a', C]]$.

Often, the analyst is unable to obtain information on all relevant confounders. In the language of causal graphs, this corresponds to the existence of unmeasured variable(s) $U$, such that structures of the form $A \leftarrow U \rightarrow Y$ are present in the underlying hidden variable causal model (i.e., $U$ is a common cause of $A$ and $Y$). Such structures are often summarized via a bidirected edge $A \leftrightarrow Y$ in graphical representations of the observed margin of the model known as *acyclic directed mixed graphs* (ADMGs). Simple covariate adjustment is insufficient to obtain unbiased estimates of the causal effect in such settings. However, Pearl [1995a] showed that if one were able to obtain measurements on a mediator (or set of mediators) $M$ such that the causal structure shown in Fig. 1(a) holds, then even if all common confounders are unobserved, the counterfactual mean is identified as the following functional of the observed data:

$$\mathbb{E}[Y \mid \mathrm{do}(a)] = \sum_M p(M|A = a) \times \sum_A p(A) \times \mathbb{E}[Y|A, M]. \quad (1)$$

Fig. 1(a) is known as the *front-door model* and the corresponding functional is called the *front-door formula*. In our motivating example, the analyst might posit hypertension as being the primary mediating variable by which smoking leads to increased risk of coronary heart disease. Though the front-door model allows for unmeasured confounding between $A$ and $Y$, it encodes 2 key assumptions

**(F1)** An exclusion restriction implying $A$ affects $Y$ only via the mediators $M$, i.e., the direct edge $A \rightarrow Y$ is absent.

**(F2)** No unmeasured confounding between the treatment-mediator and mediator-outcome pairs, i.e., the bidirected edges $A \leftrightarrow M$ and $M \leftrightarrow Y$ are absent.

It is the absence of these edges that are typically questioned in the literature – e.g, one might be concerned that the same unmeasured variable $U$ that confounds the relation between smoking and heart disease, also confounds other relations involving smoking and hypertension, or hypertension and heart disease [Koller and Friedman, 2009, Imbens, 2020].

Given information on just $A$, $M$, and $Y$, the conditions (F1) and (F2) are untestable, as the front-door model shown in Fig. 1(a) imposes no restrictions on the observed distribution. However, consider the ADMG shown in Fig. 1(b), where we incorporate information on an additional "anchor" variable $Z$. Here, $Z$ is a common cause of both the treatment and the

mediator, but does not directly cause $Y$. In our example, the analyst may hypothesize prior history of hypertension as a candidate anchor variable. While the missing edge between $Z$ and $Y$ in Fig. 1(b) does not correspond to an ordinary conditional independence (there are no independence facts implied by the model at all), it does encode a generalized equality constraint a.k.a Verma constraint. In particular, the model imposes a well-known restriction that the Markov kernel $q_Y(Y|M) \equiv \sum_A p(A|Z) \times p(Y|Z, A, M)$ is not a function of $Z$ [Robins, 1986, Verma and Pearl, 1990, Spirtes et al., 2000]. Alternatively, this constraint may be viewed as a "dormant" independence stating $Z \perp\!\!\!\perp Y$ in a re-weighted distribution $p(Z, A, M, Y)/p(M|A, Z)$ which corresponds to the post-intervention distribution $p(Z, A, Y | \mathrm{do}(m))$. Markov kernels and their relation to post-intervention distributions are discussed in more detail in Section 3.

Though different configurations of ADMGs (e.g., switching $Z \to M$ to $Z \leftrightarrow M$, or deleting the edge entirely) may yield models with the same restriction, we show that all such configurations share a common structure on the subgraph pertaining to $A, M,$ and $Y$ that satisfies the conditions (F1) and (F2). That is, any empirical test designed to check the Verma constraint (under mild assumptions formalized in Section 4), also serves to confirm whether the front-door conditions are true. Note that the identification functional for $\mathbb{E}[Y | \mathrm{do}(a)]$ in Fig. 1(b) is not precisely the same as the front-door formula in (1), but a slight generalization of it that allows for the inclusion of baseline covariates (which may be useful in many practical settings.) The theory we propose allows for testing of the front-door conditions as well as some general versions of it. However, for ease of exposition, we refer to these general versions as simply "front-door."[1] The corresponding identifying functional for $\mathbb{E}[Y | \mathrm{do}(a)]$ in Fig. 1(b) is [Tian and Pearl, 2002],

$$\sum_{Z,M} p(Z) \times p(M|a, Z) \times \sum_A p(A|Z) \times \mathbb{E}[Y|Z, A, M]. \quad (2)$$

In Section 5, we show how inverse probability weighted estimators for the above functional described by Fulcher et al. [2020] and Bhattacharya et al. [2020] can be adapted to design empirical tests for the Verma constraint and subsequent estimation of effects. Readers familiar with IV methods might wonder whether the anchor $Z$ also satisfies the IV conditions. While in the case of Fig. 1(b) it does not (the exclusion restriction that all causal paths from $Z$ to $Y$ must go through $A$ is not met), Section 6 discusses an intersection model where both IV and front-door conditions hold.

## 3  CAUSAL GRAPHICAL MODELS

The causal model of a DAG $\mathcal{G}(V)$ defined over a set of variables $V$ can be understood as the set of distributions induced

---

[1]We will briefly note how the theory trivially extends when there are additional baseline covariates $C$ besides the anchor.

by a system of structural equations – one equation for each vertex $V_i$ as a function of its "parents" $\mathrm{pa}_{\mathcal{G}}(V_i)$ and a noise term $\epsilon_i$ – equipped with the $\mathrm{do}(\cdot)$ operator [Pearl, 2009]. Typically, the noise terms in the system are assumed to be mutually independent, though this is not strictly necessary [Richardson and Robins, 2013]. The criteria we describe are non-parametric in the sense that they do not rely on any extra distributional assumptions on the structural equations or noise terms. The system induces a joint distribution $p(V)$ over the observed variables that factorizes according to $\mathcal{G}(V)$ as follows: $p(V) = \prod_{V_i \in V} p(V_i \mid \mathrm{pa}_{\mathcal{G}}(V_i))$. Further, counterfactual distributions arising from interventions on subsets of variables $A \subset V$, written as $p(V \setminus A \mid \mathrm{do}(a))$, are given by a truncated factorization, often referred to as the g-formula, where conditional factors for each $A_i \in A$ are dropped [Robins, 1986, Spirtes et al., 2000, Pearl, 2009].

$$p(V \setminus A \mid \mathrm{do}(a)) = \left. \frac{\prod_{V_i \in V} p(V_i \mid \mathrm{pa}_{\mathcal{G}}(V_i))}{\prod_{A_i \in A} p(A_i \mid \mathrm{pa}_{\mathcal{G}}(A_i))} \right|_{A=a}. \quad (3)$$

Often the analyst is unable to obtain measurements on all variables in the system. In such cases it may be inconvenient to work directly with the hidden variable causal DAG $\mathcal{G}(V \cup U)$, where $U$ is the set of unmeasured variables. A popular alternative is to use an ADMG $\mathcal{G}(V)$ consisting of directed ($\to$) and bidirected ($\leftrightarrow$) edges to model the observed data margin via a nested factorization of Markov kernels [Richardson et al., 2017]. The ADMG $\mathcal{G}(V)$ can be constructed from the DAG $\mathcal{G}(V \cup U)$ using the latent projection operation described by Verma and Pearl [1990]. A directed edge $V_i \to V_j$ in $\mathcal{G}(V)$ maintains the usual causal interpretation; a bidirected edge $V_i \leftrightarrow V_j$ can be construed (wlog) as the presence of one or more unmeasured confounders $V_i \leftarrow U_k \to V_j$ in the underlying hidden variable DAG $\mathcal{G}(V \cup U)$ [Evans, 2018]. The nested Markov factorization has the desired property that it preserves all non-parametric equality restrictions implied on the observed margin by the hidden variable DAG, and permits phrasing of causal identification algorithms on ADMGs without loss of generality [Evans, 2018, Shpitser and Pearl, 2006, Richardson et al., 2017]. We now briefly describe this factorization using conditional ADMGs (CADMGs); for a more detailed overview, see Appendix E.

A CADMG $\mathcal{G}(V, W)$ is a special kind of ADMG used to describe post-intervention distributions where variables in $V$ are random, and those in $W$ are fixed to constants via intervention. The nested Markov factorization of an ADMG $\mathcal{G}(V)$ can then be described in terms of Markov kernels of the form $q_D(D | \mathrm{pa}_{\mathcal{G}}(D))$ where each set $D$ is a subset of $V$ that forms a bidirected connected component in a CADMG $\mathcal{G}(D, V \setminus D)$ representing a post-intervention distribution where all variables in $V \setminus D$ are fixed by intervention, and this distribution is identified from $p(V)$ via sequential application of the g-formula. Such a set $D$ is said to be an *intrinsic set*.

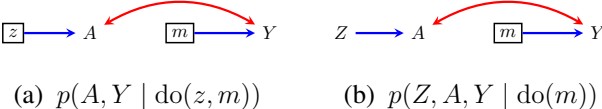

(a) $p(A, Y \mid \mathrm{do}(z, m))$      (b) $p(Z, A, Y \mid \mathrm{do}(m))$

Figure 2: Examples of CADMGs corresponding to the intervention distributions obtained from the ADMG in Fig. 1(b).

As an example, consider the ADMG in Fig. 1(b). The post-intervention distribution $p(A, Y \mid \mathrm{do}(z, m))$ is identified as $p(Z, A, M, Y)/\{p(Z) \times p(M \mid A, Z)\}$ – the g-formula can be applied to fix $Z$ first and then $M$, or vice-versa. The set $\{A, Y\}$ also forms a bidirected connected component in the corresponding CADMG shown in Fig. 2(a); thus, it is intrinsic. The associated Markov kernel is $q_{AY}(A, Y \mid Z, M) \equiv p(A \mid Z) \times p(Y \mid Z, A, M)$, i.e., the functional obtained via sequential application of the g-formula to $Z$ and $M$. Given this description, the list of all Markov kernels corresponding to intrinsic sets in Fig. 1(b) is:

$$q_Z(Z) \equiv p(Z), \tag{4}$$
$$q_A(A \mid Z) \equiv p(A \mid Z),$$
$$q_M(M \mid A, Z) \equiv p(M \mid A, Z),$$
$$q_{AY}(A, Y \mid Z, M) \equiv p(A \mid Z) \times p(Y \mid Z, A, M),$$
$$q_Y(Y \mid M) \equiv \sum_A p(A \mid Z) \times p(Y \mid Z, A, M).$$

Let $\mathcal{D}(\mathcal{G}(V, W))$ denote the set of all bidirected connected components of random variables, commonly referred to as *districts*, in the CADMG $\mathcal{G}(V, W)$. The nested Markov factorization states that the observed distribution $p(V)$ satisfies the following *district factorization* wrt to the ADMG $\mathcal{G}(V)$:

$$p(V) = \prod_{D \in \mathcal{D}(\mathcal{G})} q_D(D \mid \mathrm{pa}_{\mathcal{G}}(D)), \tag{5}$$

where each kernel appearing in this factorization corresponds to intrinsic sets in $\mathcal{G}(V)$. In Fig. 1(b), this implies: $p(V) = q_Z(Z) \times q_M(M|A, Z) \times q_{AY}(A, Y|Z, M)$. The nested factorization further asserts that any post-intervention distribution $p(V \setminus S \mid \mathrm{do}(s))$ identified from $p(V)$ satisfies the district factorization wrt to the corresponding CADMG $\mathcal{G}(V \setminus S, S)$, where again each kernel in the factorization corresponds to intrinsic sets [Richardson et al., 2017].

Ordinary independence constraints implied by the nested Markov model of $p(V)$ can be read via an extension of the well-known d-separation criterion for DAGs that extends the notion of a collider to include structures of the form $\to \circ \leftrightarrow$, $\leftrightarrow \circ \leftarrow$, and $\leftrightarrow \circ \leftrightarrow$. Generalized independence constraints a.k.a Verma constraints can also be read via m-separation applied to CADMGs corresponding to post-intervention distributions formed via multiplication of intrinsic kernels.

# 4 TESTABILITY OF FRONT-DOOR

In this section we prove a result on the testability of front-door assumptions using a generalized equality constraint between the outcome $Y$ and anchor variable $Z$. Consider the ADMG in Fig. 1(b), and the CADMG in Fig. 2(b) which corresponds to the post-intervention distribution $p(Z, A, Y | \mathrm{do}(m)) = q_Z(Z) \times q_{AY}(A, Y|Z, M = m)$ (this is derived by applying district factorization to the CADMG with intrinsic kernels defined in (4).) If we apply m-separation to the ADMG in Fig. 1(b), we detect no ordinary independence constraints between $Z$ and $Y$. However, applying m-separation to the CADMG in Fig. 2(b), we see that $Z \perp\!\!\!\perp Y$ in $p(Z, A, Y | \mathrm{do}(m))$. Alternatively, this constraint may be viewed as saying that the intrinsic kernel $q_Y(Y|M) = \sum_A q_{AY}(A, Y|Z, M)$ (compare the two kernels in (4)) is not a function of $Z$. Since intrinsic kernels always correspond to post-intervention distributions that are identified from the observed distribution, this implies a testable restriction on the observed data distribution $p(Z, A, M, Y)$. This is an example of a dormant independence a.k.a Verma constraint [Shpitser and Pearl, 2008].

Below, we formally define the concept of an anchor variable and assumptions under which the above constraint can be used to empirically verify the front-door assumptions.

**(A1)** $M$ is a mediator between $A$ and $Y$.

**(A2)** $Z$ is a covariate that is *not* a causal consequence of $A$ such that $Z \not\perp\!\!\!\perp A$ and $Z \not\perp\!\!\!\perp Y \mid A, M$.

**(A3)** A general version of *faithfulness* (Verma faithfulness) stating that all non-parametric equality restrictions in distributions $p(V)$ that nested Markov factorize wrt an ADMG $\mathcal{G}(V)$ are due to its structure (ruling out coincidental cancellations in pathways for example.) That is, an ordinary independence in $p(V)$ implies m-separation in $\mathcal{G}(V)$, and a generalized independence in a post-intervention distribution (or kernel) obtained from $p(V)$ implies (i) identifiability of this post-intervention distribution given the structure of $\mathcal{G}$ and (ii) m-separation in the corresponding CADMG.

We briefly provide justification and intuition for these assumptions (more details are in Appendix A.) (A1) simply requires that the analyst believes that $M$ in fact mediates the effect of $A$ on $Y$, but does not impose any other restrictions implied by the front-door model (e.g., absence of a direct effect of $A$ on $Y$ or absence of confounding along the pathway through $M$.) (A2) is a "relevance" assumption that is automatically satisfied if there exists either $Z \to A$ or $Z \leftrightarrow A$ (or both) in conjunction with the edge $A \leftrightarrow Y$. That is, the assumption is met when $Z$ directly affects or is confounded with $A$, and $A$ and $Y$ share an unmeasured confounder (the primary motivation for applying front-door adjustment.) We define any variable $Z$ satisfying assumption (A2) to be an anchor variable. Similar definitions of an anchor are used

by Entner et al. [2013] and Shah et al. [2022] in the context of testing validity of covariate adjustment sets. Finally, (A3) subsumes the standard faithfulness assumption employed in causal discovery methods based on ordinary independence constraints by noting that such constraints do not rely on computation of post-intervention distributions. General versions of faithfulness, similar to (A3), are used in works like Shpitser et al. [2014] and Bhattacharya et al. [2021] that incorporate Verma constraints into causal discovery.

As noted in Section 2, the criterion we propose can be used to verify the front-door conditions and generalizations of it. Specifically, Tian and Pearl [2002] showed that the causal effect of $A$ on all other variables in an ADMG $\mathcal{G}(V)$ is identified if and only if $A$ has no bidirected path to any of its children; it is easy to confirm that this criterion includes the front-door model as a special case. We now formalize a result on the testability of this condition.

**Theorem 1.** *If the generalized equality constraint $Z \perp\!\!\!\perp Y$ in $p(Z, A, M, Y)/p(M|A, Z)$ holds in some distribution $p(Z, A, M, Y)$ satisfying assumptions (A1-A3), then this distribution nested Markov factorizes wrt an ADMG where $A$ has no bidirected paths to its children.*

The intuition is as follows (see Appendix G for all proofs.) Under Verma faithfulness, any $p(Z, A, M, Y)$ satisfying the Verma constraint in Theorem 1 must be nested Markov wrt an ADMG $\mathcal{G}$ where: (i) $p(Z, A, Y | \operatorname{do}(m))$ is identified, and (ii) $Z$ and $Y$ are m-separated in the corresponding CADMG obtained by deleting incoming edges to $M$. Distributions that factorize wrt to ADMGs where $A$ has a bidirected path to one of its children are incompatible with one or both of these requirements. For example, adding $A \to Y$ to Fig. 1(b) in violation of the exclusion restriction results in a trivial bidirected path from $A$ to its child $Y$. The model implies $p(Z, A, Y | \operatorname{do}(m))$ is identified, however, $Z$ and $Y$ are not m-separated in the resulting CADMG. On the other hand, in cases where $A$ has a bidirected path to $M$, the kernel $p(Z, A, Y | \operatorname{do}(m))$ is not identified from observed data. For example, violating (F2) of the front-door criterion by adding $A \leftrightarrow M$ or $M \leftrightarrow Y$ to Fig. 1(b) leads to graphs where $M$ also has a bidirected path to its child $Y$, which results in non-identification per Tian and Pearl [2002]. A pattern representation of all ADMGs that satisfy the Verma constraint is shown in Fig. 3(a). In the pattern, presence of solid edges between $A, M, Y$ and absence of any other edges between them correspond to the front-door model, and is "compelled" by the Verma constraint; the dashed edges can be present or absent, with the restriction that at least either $Z \to A$ or $Z \leftrightarrow A$ (or both) exists (assumption A1) and $A$ has no bidirected path to $M$ (compelled by the constraint.) For an exhaustive list of valid ADMGs drawn from this pattern see Appendix B. Invalid ADMGs drawn from Fig. 3(a) are ones where $Z, A, M, Y$ form a single district; a pattern representation of these is shown in Fig.3(b).

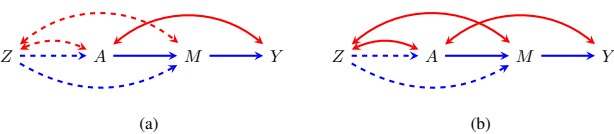

Figure 3: (a) A pattern representing ADMGs that satisfy the restriction $Z \perp\!\!\!\perp Y$ in $p(Z, A, M, Y)/p(M|A, Z)$; (b) A pattern representing ADMGs that imply no non-parametric equality constraints, and should *not* be construed from (a).

Importantly for downstream causal inference, we show that all ADMGs derived from the pattern in Fig. 3(a) share the same identification theory for the effect of $A$ on $Y$. Since $A$ has no bidirected path to its children in any such $\mathcal{G}$, the post-intervention distribution $p(Z, M, Y | \operatorname{do}(a))$ is given by a truncated version of the district factorization where we divide by a *nested* propensity score for $A$ [Tian and Pearl, 2002, Bhattacharya et al., 2020]. Let $q_{D_A}(D_A | \operatorname{pa}_{\mathcal{G}}(D_A))$ represent the intrinsic kernel corresponding to the district containing $A$ in $\mathcal{G}$. From the pattern, $D_A$ is either $\{A, Z, Y\}$ or $\{A, Y\}$. The required nested propensity score $\widetilde{q}(A|Y, Z, M)$ is derived from this kernel via conditioning on all elements in $D_A$ besides $A$. That is, $\widetilde{q}(A|Y, Z, M) = q_{D_A} / \sum_A q_{D_A}$. In the case when $D_A = \{A, Z, Y\}$ we get $\frac{p(A|Z) \times p(Y|A,Z,M)}{\sum_A p(A|Z) \times p(Y|A,Z,M)}$. It is easy to confirm from (4) that when $D_A = \{A, Y\}$ we get the same result. Based on these observations we have the following identification result wrt the patterns in Fig. 3(a).

**Lemma 1.** *In joint distributions that nested factorize wrt valid ADMGs derived from Fig. 3(a), we have $p(Z, M, Y | \operatorname{do}(a)) = p(Z, A, M, Y)/\widetilde{q}(A|Y, Z, M)|_{A=a}$, where $\widetilde{q}(A|Y, Z, M) = \frac{p(A|Z) \times p(Y|A,Z,M)}{\sum_A p(A|Z) \times p(Y|A,Z,M)}$. Since the entire post-intervention is identified, the target $\mathbb{E}[Y | \operatorname{do}(a)]$ is also identified as $\sum_{Z,M,Y} p(Z, M, Y | \operatorname{do}(a)) \times Y$.*

The above functional resembles a truncated factorization in the sense that we divide the joint $p(Z, A, M, Y)$ by a conditional kernel $\widetilde{q}(.)$ of $A$ similar to how in a fully observed DAG, intervention on $A$ would entail division by a simple conditional factor of $A$, see (3). This informs the design of tests in the next section. Jaber et al. [2019] propose general identification results based on patterns of ordinary Markov equivalence; Lemma 1 differs in that it is based on a pattern of nested Markov equivalence. Such results will become increasingly important as more causal discovery procedures that incorporate Verma constraints[Shpitser et al., 2014, Bhattacharya et al., 2021] are developed.

We end this section by noting that while the criterion in Theorem 1 is sufficient to guarantee identification via the above functional, it is not necessary. That is, verifying the presence of the Verma constraint assures the analyst that the ACE is computed in an identified model. However, situations in which the constraint does not hold fall into two cases: mod-

els where the effect is not identified, and ones in which it is, but there is no constraint between the anchor and outcome because of, say, a direct effect of $Z$ on $Y$ or confounded dependence between them. We have already discussed the former cases; as a simple example of the latter, consider the ADMG in Fig. 1(b) and add the $Z \to Y$ edge. There is no longer any Verma constraint present (the nested Markov model of this ADMG imposes no non-parametric equality restrictions whatsoever), but the identification conditions still hold. Nonetheless, the criterion is a useful pre-test for front-door adjustment and its extensions.

# 5   TESTING AND EFFECT ESTIMATION

We now discuss procedures for testing the Verma constraint and estimating the effect from finite samples. Directly testing whether the kernel $q_Y(Y|M) \equiv \sum_A p(A|Z) \times p(Y|A, M, Z)$ is not a function of $Z$ using natural parameterizations of the observed data likelihood leads to the g-null paradox [Robins and Wasserman, 1997]. Hence, we borrow ideas from inverse probability weighting (IPW) and marginal structural models [Robins, 2000] for this purpose. As mentioned in the introduction, we will propose two distinct ways of testing the constraint and also discuss non-parametric extensions of these tests.

**Primal test and Primal IPW**:

The first test is based on weights used in the primal IPW estimator for the front-door functional proposed in Bhattacharya et al. [2020]. Consider a chain factorization of the observed data $p(Z, A, M, Y) = p(Z) \times p(A|Z) \times p(M|A, Z) \times p(Y|Z, A, M)$ for any valid ADMG derived from Fig 3(a). Given this factorization, the post-intervention distribution after intervening on $A$ is identified per Lemma 1 as,

$$p(Z, M, Y \,|\, \mathrm{do}(a)) = p(Z, A, M, Y)/\widetilde{q}(A|Y, Z, M)|_{A=a} \quad (6)$$
$$= p(Z) \times p(M|a, Z) \times \sum_A p(A|Z) \times p(Y|Z, A, M).$$

This post-intervention distribution is nested Markov equivalent to the CADMG in Fig. 4(a), where we see the Verma constraint: $Y \perp\!\!\!\perp Z | M$. Testing this independence in $p(Z, M, Y \,|\, \mathrm{do}(a))$ is equivalent to testing $q_Y(Y|M)$ is not a function of $Z$ due to the following district factorization of the CADMG in terms of intrinsic kernels: $q_Z(Z) \times q_M(M|a, Z) \times q_Y(Y|M)$. That is, the independence found via m-separation in the CADMG corresponds to the same restriction that the kernel $q_Y(Y|M)$ is not a function of $Z$. To test this constraint, we need to compare the conditional kernels of $q_Y(Y|Z, M)$ and $q_Y(Y|M)$. From a causal perspective, this can be viewed as evaluating the goodness-of-fit of models $Y|Z, M$ and $Y|M$ wrt the post-intervention distribution $p(Y, Z, M \,|\, \mathrm{do}(a))$. We use ideas from marginal structural models, where causal parameters are estimated using inverse weights based on the propensity score of the treatment. Here, we can use weights de-

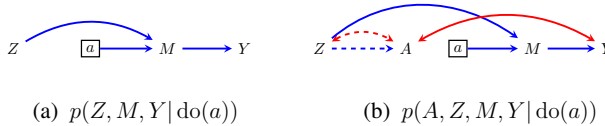

(a) $p(Z, M, Y \,|\, \mathrm{do}(a))$      (b) $p(A, Z, M, Y \,|\, \mathrm{do}(a))$

Figure 4: (a) CADMG corresponding to $p(Z, M, Y \,|\, \mathrm{do}(a))$; (b) CADMG corresponding to $p(Z, M, Y, A \,|\, \mathrm{do}(a))$.

rived from the nested propensity score of the treatment $\widetilde{q}(A|Y, Z, M)$ given its relation to the post-intervention distribution $p(Z, M, Y \,|\, \mathrm{do}(a))$ per Lemma 1. Following Bhattacharya et al. [2020], we refer to these as *primal weights*.

Formally, let $\pi(Z, M; \alpha_y) \coloneqq p(Y|Z, M, \mathrm{do}(a); \alpha_y)$ and $\pi(M; \beta_y) \coloneqq p(Y|M, \mathrm{do}(a); \beta_y)$, where $\alpha_y$ and $\beta_y$ denote the set of parameters used to model the corresponding distributions. We can consistently estimate $\alpha_y$ and $\beta_y$ using samples from the observed distribution $p(Z, A, M, Y)$ via the following unbiased estimating equations:

$$\mathbb{P}_n \left[ \frac{U(\pi(Z, M; \alpha_y))}{\widetilde{q}(A \mid Y, Z, M; \widehat{\eta})} \right] = 0, \mathbb{P}_n \left[ \frac{U(\pi(M; \beta_y))}{\widetilde{q}(A \mid Y, Z, M; \widehat{\eta})} \right] = 0, \quad (7)$$

where $\mathbb{P}_n[.] \coloneqq \frac{1}{n} \sum_{i=1}^n (.); \mathbb{P}_n[U(\pi(Z, M; \alpha_y))] = 0$ and $\mathbb{P}_n[U(\pi(M; \beta_y))] = 0$ are unbiased estimating equations for $\alpha_y$ and $\beta_y$ under the observed distributions $p(Y|Z, M)$ and $p(Y|M)$, respectively; $\widehat{\eta}$ denotes the estimated parameters for $p(A|Z)$ and $p(Y|Z, A, M)$ used to compute primal weights $1/\widetilde{q}(A|Y, Z, M)$. Once $\alpha_y$ and $\beta_y$ are estimated, we can compare goodness-of-fit between $\pi(Z, M; \alpha_y)$ and $\pi(M; \beta_y)$ via likelihood ratio or Wald tests (the latter only requires $\alpha_y$) [Robins and Wasserman, 1997, Agostinelli and Markatou, 2001]. The procedure can be summarized as:

1. Fit models for $p(A|Z)$ and $p(Y|Z, A, M)$, and predict primal weights $1/\widetilde{q}(A|Y, Z, M)$ for each row of data,

2. Use the estimated weights to fit weighted regressions $p(Y|Z, M, \mathrm{do}(a))$ and $p(Y|M, \mathrm{do}(a))$ using (7), and compare goodness of fits between these two models.

If the test indicates that the Verma constraint holds with some pre-specified significance level, this suggests we are in a model given by the equivalence class in Fig. 3(a), and the effect is identified. We can then re-use the models fitted above to compute the counterfactual mean using the primal IPW estimator proposed in Bhattacharya et al. [2020]:

$$\widehat{\mathbb{E}}[Y \,|\, \mathrm{do}(a)] = \mathbb{P}_n \left[ \frac{\mathbb{I}(A = a)}{\widetilde{q}(A \mid Y, Z, M; \widehat{\eta})} \times Y \right]. \quad (8)$$

**Dual test and Dual IPW**:

The Verma test based on primal weights relies on correct specification of the treatment and outcome models. We now provide alternatives that instead rely on specification of the mediator model $p(M|A, Z)$. The post-intervention distribution after intervening on $M$ in any valid

ADMG derived from the pattern in Fig 3(a) is identified as: $p(Z, A, Y | \operatorname{do}(m)) = p(Z) \times p(A|Z) \times p(Y|Z, A, M = m)$, which is nested Markov equivalent to the CADMG in Fig. 2(b); here we see the usual phrasing of the Verma constraint $Z \perp\!\!\!\perp Y$ in $p(Z, A, Y | \operatorname{do}(m))$. One way to empirically test the constraint is to use a similar procedure as the one described with the primal weights, but instead compare goodness-of-fit for $p(Y|Z, \operatorname{do}(m))$ and $p(Y | \operatorname{do}(m))$ using inverse weights $1/p(M|A, Z)$; this is the g-null test described in Robins [1986], Robins and Wasserman [1997]. However, typical IPW weights may suffer from various numerical issues, so instead we describe a stabilized version of the g-null test that uses weights which can also be plugged into the dual IPW estimator for $\mathbb{E}[Y | \operatorname{do}(a)]$ proposed by Bhattacharya et al. [2020]. The *dual weights* use a ratio of densities (which leads to stabilization of weights) as follows:

$$q^d(M|A, Z) \equiv \frac{p(M|A, Z)}{p(M|A = a, Z)}, \quad (9)$$

for any given choice of intervention value $A = a$.

The reason these weights are suitable for this purpose is due to its relation to the following post-intervention distribution:

$$p(A, Z, M, Y | \operatorname{do}(a)) = p(Z, A, M, Y)/q^d(M|A, Z).$$

This post-intervention distribution is nested Markov wrt the CADMG shown in Fig. 4(b) where both fixed $a$ and random $A$ are present. Such CADMGs arise in single world intervention graph (SWIG) interpretations of identification algorithms [Bhattacharya et al., 2020, Shpitser et al., 2020]. It can be confirmed that $p(Z, M, Y | \operatorname{do}(a))$ is obtained by simply marginalizing over $A$ in the above equation. Similar to the CADMG in Fig. 4(a) we have $Z \perp\!\!\!\perp Y|M$ in Fig. 4(b) corresponding to $q_Y(Y|M)$ not being a function of $Z$. A two-step testing procedure can be summarized as follows:

1. Fit a model for $p(M|A, Z)$ and predict dual weights $1/q^d(M|A, Z)$ for each row of data.[2]

2. Use the estimated weights to fit weighted regressions $p(Y|Z, M, \operatorname{do}(a))$ and $p(Y|M, \operatorname{do}(a))$ using (7), but with $q^d(M|A, Z)$ in the denominator, and compare goodness-of-fit between these two models.

If the test succeeds, we can re-use the same models in the following dual IPW estimator [Bhattacharya et al., 2020]:

$$\widehat{\mathbb{E}}[Y | \operatorname{do}(a)] = \mathbb{P}_n \left[ \frac{p(M|A = a, Z; \widehat{\eta})}{p(M | A, Z; \widehat{\eta})} \times Y \right]. \quad (10)$$

**Non-parametric extensions of primal and dual tests**:

Given any non-parametric test $\tau(Y, Z, M)$ that is appropriate for testing an ordinary independence $Y \perp\!\!\!\perp Z|M$,

---

[2]This step may also be improved using ideas in Menon and Ong [2016] for estimating density ratios directly.

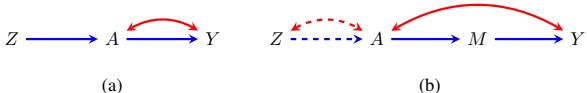

Figure 5: (a) The classical instrumental variable model; (b) The front-door model with an instrumental variable $Z$.

Thams et al. [2021] propose to test the generalized constraint $Y \perp\!\!\!\perp Z|M$ in $p(Z, M, Y | \operatorname{do}(a))$ by applying $\tau$ to a pseudo-dataset that mimics this post-intervention distribution. This pseudo-dataset is created via a resampling scheme where each row is resampled with some (potentially unnormalized) probability $1/p(A | \cdot)$, where $p(A | \cdot)$ corresponds to the propensity score required to obtain the post-intervention distribution where independence holds. That is, the resampling is done based on the usual inverse probability weights used to estimate the effect of $A$ on any downstream outcomes. While the propensity scores in Thams et al. [2021] corresponded to simple conditional distributions as in a conditionally ignorable model, this technique can be directly adapted to our methods by resampling the pseudo-dataset based on the nested propensity score (primal weights) or the dual weights. In our experiments we design a non-parametric test by applying the Fast Conditional Independence Test [Chalupka et al., 2018] in pseudo-datasets created via sampling with dual weights estimated via random forests rather than parametric models. A more detailed explanation is provided in Appendix C.

# 6 INTERSECTION WITH IV MODELS

Consider the subpattern in Fig. 5(b) corresponding to the ADMGs in Fig 3(a) that do not include any edge between $Z$ and $M$. Since these ADMGs are consistent with the pattern in Fig. 3(a) they still satisfy the front-door conditions (in fact, these correspond to the classical front-door assumptions in Pearl [1995a]) and imply the Verma restriction discussed in previous sections. In addition, $Z$ also satisfies the instrumental variable condition in these graphs. A variable $Z$ is said to satisfy the IV conditions wrt $A$ and $Y$ in $\mathcal{G}$ if (the following applies to the "classical" IV model – for more general definitions, see van der Zander et al. [2015]):

**(I1)** $Z \rightarrow A$ or $Z \leftrightarrow A$ or both exist in $\mathcal{G}$.

**(I2)** $Z$ and $Y$ are m-separated in a sub-graph where $A$ and edges involving $A$ are deleted.

ADMGs where the additional IV assumptions hold are easily distinguished from other valid ADMGs in Fig. 3(a) by noting that they encode an additional ordinary independence constraint: $Z \perp\!\!\!\perp M|A$. This leads to a simple corollary.

**Corollary 1.1.** *Under assumptions (A1-A3), distributions $p(Z, A, M, Y)$ that satisfy both $Z \perp\!\!\!\perp Y$ in $p(Z, A, M, Y)/p(M|A, Z)$ and $Z \perp\!\!\!\perp M|A$ nested Markov*

*factorize wrt an ADMG satisfying the front-door conditions (F1) and (F2), and IV conditions (I1) and (I2).*

If conditions (I1), (I2), and a third condition usually phrased as some form of effect homogeneity (e.g., absence of effect modification due to unmeasured variables $U$; see Hernán and Robins [2010] for other examples) are satisfied for a binary instrument $Z$ and binary treatment $A$, then $\mathbb{E}[Y \mid \mathrm{do}(a = 1)] - \mathbb{E}[Y \mid \mathrm{do}(a = 0)]$ is identified as $\{\mathbb{E}[Y|Z = 1] - \mathbb{E}[Y|Z = 0]\}/\{\mathbb{E}[A|Z = 1] - \mathbb{E}[A|Z = 0]\}$. Though the IV estimated effect requires additional restrictions beyond structural assumptions encoded in the graph, it would be interesting to explore in future work how estimates from the IV and front-door assumptions can be combined to obtain robustness against misspecification in either model.

# 7 EXPERIMENTS

The experiments focus on 3 tasks: (i) Studying effectiveness of the primal and dual weights for testing front-door assumptions via Verma constraints; (ii) Comparing effect estimates using front-door and IV adjustment when both assumptions hold and when only front-door assumptions hold; (iii) Demonstrating use of our methods in real-world analyses related to the motivating example in Section 2. Explicit descriptions of all simulated ADMGs and corresponding data generating processes can be found in Appendix F. Python code for our methods can be found at `https://github.com/rbhatta8/fdt`.

**Task (i)**: We consider hidden variable causal models whose observed margins $p(Z, A, M, Y)$ nested factorize wrt 4 different ADMGs: two from Fig. 3(a) in which the Verma constraint and front-door assumptions hold, and two where the assumptions do not hold due to additional confounding $A \leftrightarrow M$ and $M \leftrightarrow Y$, or violation of the exclusion restriction with $A \rightarrow Y$. We run 200 trials of the following experiment at sample sizes ranging from 200 to 20000. In a given trial we generate data from one of the four ADMGs picked at random, and compute p-values for the Verma constraint using the primal test, dual test, and Fast Conditional Independence test with dual weights fit via random forests[3] as described in Section 5. We use each method's p-values at a significance level $\alpha = 0.05$ to accept/reject the null hypothesis of a model where the Verma constraint and front-door assumptions hold. We then compute true positive and false positive rates as shown in Fig. 6. All methods quickly achieve true positive rates of $\sim 95\%$ or higher reflecting that type I error (falsely rejecting the null) is controlled at the desired significance level. False positive rates also drop asymptotically with more samples. The non-parametric test, whose performance is captured by the lines corresponding

---

[3]The non-parametric test is evaluated with data sets where the relations between variables are non-linear.

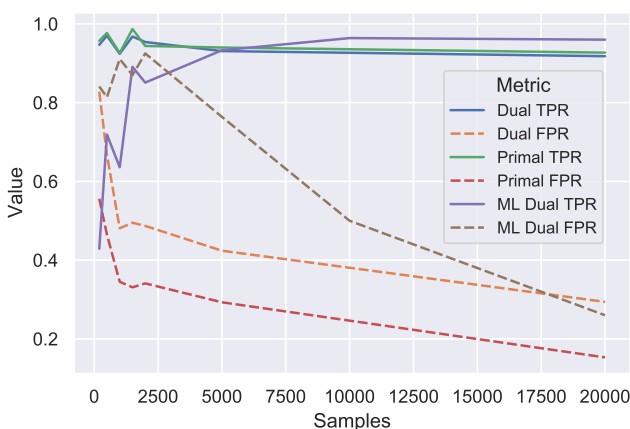

Figure 6: Comparison of different testing procedures.

to ML Dual TPR and ML Dual FPR in Fig. 6, is unstable at low sample sizes, but significantly improves with more samples. The primal test outperforms the dual test; whether this is an empirical observation or has theoretical justification is an interesting question for future work. The average bias in downstream causal effect estimates in true positive scenarios (via primal or dual tests) is only $0.04$ at a sample size of $5000$ compared to $0.28$ in false positive scenarios, highlighting the importance of accurate pre-tests.

**Task (ii)**: We generate data from one ADMG in which both the front-door and IV assumptions hold and one in which only the front-door assumptions hold. We then compute causal effect estimates using primal IPW, dual IPW, and IV adjustment. In the former case, all methods give unbiased estimates, but IV estimates have higher variance (Fig. 7(a).) In the latter case, primal and dual IPW remain unbiased, while IV adjustment is significantly biased (Fig. 7(b).) This raises a question of whether semiparametric estimators can combine all 3 methods to improve statistical efficiency, and provide robustness against misspecification of not just statistical models, but also different identifying assumptions.

**Task (iii)**: For the final task here, we analyze the effect of smoking (treatment $A$) on developing coronary heart disease (outcome $Y$) using data from the Framingam heart study [Kannel and Gordon, 1968]. Following Section 2, we propose hypertension as a candidate mediator $M$ and past history of hypertension as an anchor $Z$. We also include baseline covariates $C$ containing *age*, *sex*, *BMI*, and *past history of heart disease*. The influence of $C$ on $Z, A, M, Y$ can be easily incorporated in our framework by noting that the Verma constraint is now a dormant *conditional* independence: $Z \perp\!\!\!\perp Y | C$ in $p(C, Z, A, Y | \mathrm{do}(m))$. All densities/regressions are adapted accordingly to include $C$ in the conditioning set, e.g., we would use $q^d \equiv p(M|A, Z, C)/p(M|A = a, Z, C)$ to fit causal parameters for $p(Y|Z, M, C, \mathrm{do}(a))$ and $p(Y|M, C \,\mathrm{do}(a))$ in the dual Verma test. More details on including baseline covariates

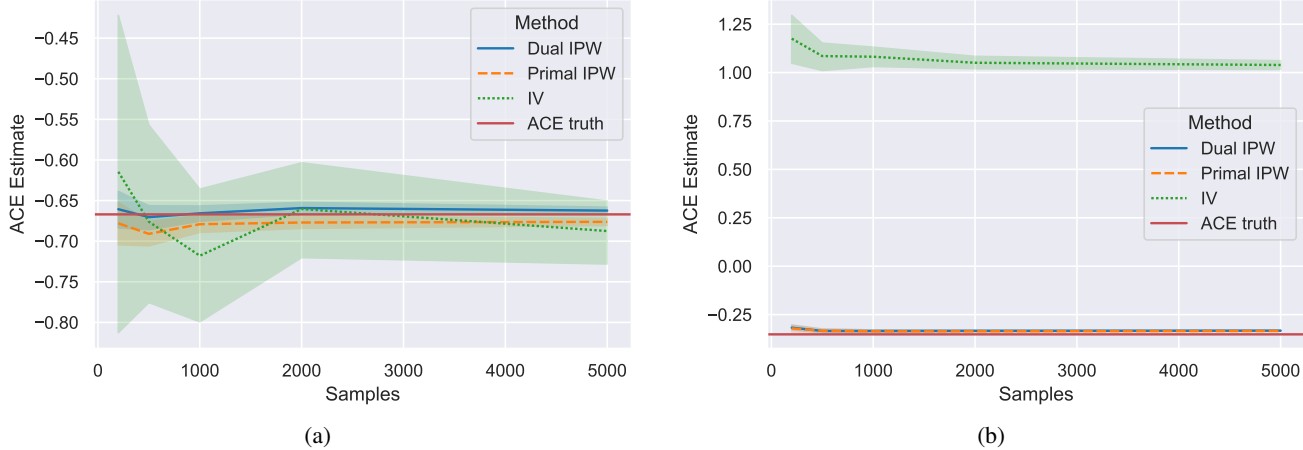

Figure 7: (a) Comparison of front-door and IV effect estimates when both assumptions hold; (b) Comparison of front-door and IV effect estimates when only front-door assumptions hold.

are in Appendix D.

For modeling flexibility, we apply the non-parametric test with dual weights. We also apply non-parametric tests for $Z \perp\!\!\!\perp M \mid C, A$ to check if $Z$ is a valid (conditional) IV, and $Z \perp\!\!\!\perp Y \mid C, A$; the latter conditional independence is an anchor variable based criterion proposed by Entner et al. [2013] to test if a set of covariates $C$ satisfies the backdoor criterion. As shown in Table 1, only the test for front-door assumptions succeeds (with $\alpha = 0.05$.) The corresponding point estimate and $95\%$ confidence intervals (using 200 bootstraps) suggest that any amount of smoking (vs. complete abstention) slightly increases the risk of heart disease – $A$ and $Y$ are encoded as binary variables in the data, so these numbers correspond to $p(Y \mid \mathrm{do}(a = 1)) - p(Y \mid \mathrm{do}(a = 0))$.

Table 1: Results from the Framingham heart study analysis.

| Method | p-value | Effect estimate |
|--------|---------|-----------------|
| Front-door | 0.5 | 0.014 (0.005, 0.021) |
| IV | 0.005 | Not applicable |
| Back-door | 0.007 | Not applicable |

## 8   CONCLUSION

Based on a testable generalized equality constraint, we have proposed ways to pre-test the front-door model and its extensions. These tests rely on variationally independence pieces of the observed data likelihood. Bhattacharya et al. [2020] have designed doubly robust semiparametric estimators for the average causal effect in these scenarios – a direction for future work is to investigate whether the pre-tests them-

selves can be made doubly robust. We have also proposed scenarios in which both the front-door and IV assumptions hold, which we hope leads to future work on combining estimates across the two models to gain additional robustness.

**Author Contributions**

RB conceived the original idea. RB and RN contributed to development of the proposed framework. RB performed the statistical analysis. RB and RN contributed to the write up and revision of the paper.

**Acknowledgements**

We thank the anonymous reviewers for their insightful comments which improved the presentation of the paper.

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
