# OpenReview forum: "On Testability of the Front-Door Model via Verma Constraints"
_auai.org/UAI/2022/Conference — UAI 2022 Poster_

### Official Review · Reviewer_qTy4 · 2022-03-30

**Q2(1) Originality/Novelty:** 3
**Q2(2) Significance/Impact:** 3
**Q2(3) Correctness/Technical Quality:** 3
**Q2(6) Clarity Of Writing:** 2
**Q6 Overall Score:** 7
**Q8 Confidence In Your Score:** 2

**Q1 Summary And Contributions:**

The authors propose a way to test that the assumptions of the front-door model are satisfied. For this they use a Verma constraint between an anchor variable and the outcome. Moreover, the authors propose finite sample tests, justify them theoretically and empirically validate them in simulation studies.Finally, they compare and in some sense combine their result with known results for IV.

**Q2 Assessment Of The Paper:**

More detailed information regarding each of these aspects is given below:

**Q2(4) Quality Of Experiments (Optional):**

3: Good: The experimental evaluation is adequate, and the results convincingly support the main claims.

**Q2(5) Reproducibility:**

3: Good: Key resources (e.g., proofs, code, data) are available and key details (e.g., proofs, experimental setup) are sufficiently well-described for competent researchers to confidently reproduce the main results.

**Q3 Main Strengths:**

- I think this paper adresses a very relevant issue in practice and also hints at several future ideas that seem well worth exploring.
- The authors do not stop after developing necessary theoretical concepts (e.g. the appropriate Verma constraint), but go a long way into the direction of practical applications by providing several finite sample tests and additional information on the handling of further covariates.

**Q4 Main Weakness:**

It might be just me, but although the paper seems written in a very precise and careful way, it was easy for me (not having a strong background on Verma constraints and the handling of CADMGs) to get lost in technicalities and loose the big picture. This is a pitty, since I thought the main message was very interesting and relevant but I could not digest the details to really appreciate it. Depending on the intended audience, of course, this might not pose a problem at all.

**Q5 Detailed Comments To The Authors:**

I do not have many detailed comments, as I think the paper was written with great care towards the details - I did not find any relevant issues.

My main suggestion would be to think how this very interesting result could be made more accessible to a wider audience. Here are some thoughts:
- I had problems following the material from the time CADMGs and Markov kernels corresponding to intrinsic sets were introduced. It might have helped if the identification of intrinsic sets in the example of Fig. 1(b) would be more explicit (e.g. are singletons also intrinsic sets?)
- eq (7): I do not find a definition of U(.)
- Make the distinction between the three testing methods clearer, e.g. by introducing bold face headings with names that are easily recognized in the figures in the experiment secion (e.g. "primal test", "dual test", "ML dual"); in Fig. 6a I understand what "primal" and "dual" is referring to, but I only guess that "ML dual" is referring to the non-parametric test sketched at the end of chapter 5 and in Appendix C.

**Q7 Justification For Your Score:**

I like this paper because of the treatment of the problem goes the full way from theory to practical considerations. Moreover, it seems to address a quite relevant issue in practice and is full of ideas on how to extend and relate the result to other approaches.
As I indicated above, I think the material is not yet too approachable to non-experts - which is a pitty, since people from many fields might be interested in the result. Still, the results seems important and should get known.

**Q9 Complying With Reviewing Instructions:**

1: Yes.

---

### Official Review · Reviewer_tk46 · 2022-04-11

**Q2(1) Originality/Novelty:** 3
**Q2(2) Significance/Impact:** 2
**Q2(3) Correctness/Technical Quality:** 3
**Q2(6) Clarity Of Writing:** 4
**Q6 Overall Score:** 8
**Q8 Confidence In Your Score:** 2

**Q1 Summary And Contributions:**

This paper considers the task of empirically testing the assumption of the front-door model. The authors derive a sufficient graphical criterion for fulfillment of these assumptions based on a Verma constraint involving an "anchor variable" and devise corresponding finite sample tests using ideas from IWP. They also consider the intersection with instrumental variable models. Numerical experiments support the theoretical results and a small real-world application is given.

**Q10 Ethical Concerns (Optional):**

No ethical concerns.

**Q2 Assessment Of The Paper:**

More detailed information regarding each of these aspects is given below:

**Q2(4) Quality Of Experiments (Optional):**

3: Good: The experimental evaluation is adequate, and the results convincingly support the main claims.

**Q2(5) Reproducibility:**

4: Excellent: Key resources (e.g., proofs, code, data) are available and key details (e.g., proof sketches, experimental setup) are comprehensively described for competent researchers to confidently and easily reproduce the main results.

**Q3 Main Strengths:**

The paper is exceptionally well written, it is a pleasant read and very precise. The proposed test of the front-door model by means of a Verma constraint is, to my knowledge, novel and innovative. I think it constitutes a nice bridge from theory to practice and is useful for practitioners of causal inference. The results seem technically solid, although I did not check the proofs. The experiments support the central claims and it is nice to see a small real-world example. Code is provided in the supplementary material and many further details are given in the appendix.

**Q4 Main Weakness:**

The paper is quite dense and technically difficult in some places. Thus, despite being written very well, I think it can be a though read without much prior knowledge.

**Q5 Detailed Comments To The Authors:**

No major comments.


Minor comments:

- Potential TeX issue: In the equations labeled by (F1), (F2), (A1), (A2), (A3), (I1), and (I2) these labels extend beyond the left border of the textbox.

- "A directed edge $V_i \rightarrow V_j$ in $\mathcal{G}(V)$ maintains the usual causal interpretation; a bidirected edge $V_i \leftrightarrow V_j$  can be construed (wlog) as the presence of one or more unmeasured confounders $V_i \leftarrow U \rightarrow V_j$ in the underlying hidden variable DAG $\mathcal{G}(V \cup U)$": There is also an edge $V_i \rightarrow V_j$ in $\mathcal{G}(V)$ if $V_i \rightarrow U \rightarrow V_j$ in $\mathcal{G}(V \cup U)$, for example, and an edge $V_i \leftrightarrow V_j$ in $\mathcal{G}(V)$ if $V_i \leftarrow U_1 \rightarrow U_2 \rightarrow V_j$ in $\mathcal{G}(V \cup U)$, for example. To me it was clear that you also meant such cases, but to avoid any possibility for confusion you might want to consider adding a small explanatory note.

- The paragraph on CADMGs and Markov kernels in section 3 is very dense and, in my point of view, hard to understand without much prior knowledge. A few suggestions:

1. Specifically define how the CADMG $\mathcal{G}(V, W)$ is defined, i.e., by removing the edges into $W$.
2. The Markov kernels $q_D(D|\text{pa}_{\mathcal{G}}(D))$ are left undefined. Please add their definition.

- Assumption (A2): Is the "such that" to be read as "and"?

- "Finally, (A3) reduces to the standard faithfulness assumption": Do you mean that (A3) subsumes standard faithfulness? The term "reduces" to me more reads like saying that (A3) and standard faithfullness are the same.

- Theorem 1: I would suggest writing "then this distribution" instead of "then the distribution".

- "It is easy to confirm from (4) ...": --> eq. (4)".

- "Such results will become increasingly important as more causal discovery procedures that incorporate Verma constraints are developed": Please add a reference to such algorithms.

- "as a simple example of the latter, consider the ADMG in Fig. 3(b) and add the $Z \rightarrow Y$ edge. There is no longer any Verma constraint present (the nested Markov model of this ADMG imposes no non-parametric equality restrictions whatsoever), but the identification conditions still hold": Did you perhaps mean Fig. 3(a)? In the graph in Fig. 3(b) there is the bidirected path $A \leftrightarrow Z \leftrightarrow M$ from $A$ to its child $M$. So the effect of $A$ on $Y$ should as of Tian and Pearl, 2002 be non-identified. Or did I get something wrong here?

- Equation (7) and corresponding paragraph: The quantities $\alpha_y$ and $\beta_y$ are undefined.

- "Once $\alpha_y$ and $\beta_y$ are estimates": --> estimated

- "using (7)": --> eq. (7)

- Figure 6(a): I assume the labels "ML Dual" refer to the "Fast Conditional Independence test with dual weights fit via random forests". Please make this more apparent with an alternative label name.

**Q7 Justification For Your Score:**

The strengths strongly outweigh the weaknesses. Technically solid and excellently written paper with a relevant contribution to the field that seems to be useful in practice.

--------

Thanks to the authors for their answer to the reviewers' comments. I appreciate the author's efforts to improve the accessibility of the paper in its final version. I keep my positive score.

**Q9 Complying With Reviewing Instructions:**

1: Yes.

---

### Official Review · Reviewer_YP7R · 2022-04-12

**Q2(1) Originality/Novelty:** 3
**Q2(2) Significance/Impact:** 2
**Q2(3) Correctness/Technical Quality:** 3
**Q2(6) Clarity Of Writing:** 3
**Q6 Overall Score:** 7
**Q8 Confidence In Your Score:** 3

**Q1 Summary And Contributions:**

This paper proposes a way to test the front-door conditions on data by testing whether a specific Verma constraint holds in the data. Statistical tests and estimators are developed, and evaluated on synthetic as well as real-world data.

**Q2 Assessment Of The Paper:**

More detailed information regarding each of these aspects is given below:

**Q2(4) Quality Of Experiments (Optional):**

3: Good: The experimental evaluation is adequate, and the results convincingly support the main claims.

**Q2(5) Reproducibility:**

3: Good: Key resources (e.g., proofs, code, data) are available and key details (e.g., proofs, experimental setup) are sufficiently well-described for competent researchers to confidently reproduce the main results.

**Q3 Main Strengths:**

Interesting, theoretically solid paper with good potential for use in practice.

**Q4 Main Weakness:**

The technical parts can be hard to follow, though I think not so much an issue with the presentation of this paper, but rather with the amount of (often unpublished) material that is referenced. This could limit the potential audience.

**Q5 Detailed Comments To The Authors:**

Two comments concerning the phrasing of assumption (A3):
* There is no graph in the text that "the graph" could refer to. I believe this should say something along the lines of "there exists a graph such that ..."
* The two uses of "corresponding" make the text harder to follow, especially for readers not very familiar with nested Markov factorizations. Maybe this can be clarified using math notation from section 3.

**Q7 Justification For Your Score:**

My concern about the accessibility of this paper was the reason I didn't give a higher score.

I was not able to review section 5 properly due to lack of familiarity with earlier results that section builds on.

---

I would like to thank the reviewers for their response and their effort to address the concerns of accessibility. I've raised my score from 6 to 7.

**Q9 Complying With Reviewing Instructions:**

1: Yes.

---

### Official Review · Reviewer_fV8R · 2022-04-13

**Q2(1) Originality/Novelty:** 3
**Q2(2) Significance/Impact:** 2
**Q2(3) Correctness/Technical Quality:** 3
**Q2(6) Clarity Of Writing:** 2
**Q6 Overall Score:** 6
**Q8 Confidence In Your Score:** 3

**Q1 Summary And Contributions:**

The applicability of the front-door criterion for causal analysis is reduced here to a test of the classical Verma constraints. The result (Th1) gives a sufficient but not necessary condition. A statistical procedure to test such constraints is discussed. Experiments on small (artificial and real-world) models are promising.

**Q10 Ethical Concerns (Optional):**

Nothing.

**Q2 Assessment Of The Paper:**

More detailed information regarding each of these aspects is given below:

**Q2(4) Quality Of Experiments (Optional):**

3: Good: The experimental evaluation is adequate, and the results convincingly support the main claims.

**Q2(5) Reproducibility:**

2: Fair: Key resources (e.g., proofs, code, data) are unavailable but key details (e.g., proof sketches, experimental setup) are sufficiently well-described for an expert to confidently reproduce the main results.

**Q3 Main Strengths:**

The paper is clearly providing a new sufficient condition for the applicability of the front-door criterion.

**Q4 Main Weakness:**

I have found the presentation of the technical material a bit hard to grasp. My current impression is that the result derived by the author is sound and useful for the community, but the presentation is not helping the reader in getting it.





**Q5 Detailed Comments To The Authors:**

As said, I have an overall positive opinion about the paper, but I have some problem with the presentation. In particular, the authors seems to focus on a specific graph for some derivations/equations, while providing general equations in some other cases. I see that most of the ideas derived for the specific model can be easily extended, but this makes the discussion a bit confused. Besides that the statement of both the theorem and the lemma can be probably made more clear and some basic concepts (such as Markov nested factorization) could deserve a formal definition.  Similarly the testing part could be better shaped as an algorithm. Some effort to increase the reproducibility of the experiments could be also done.

**Q7 Justification For Your Score:**

As said I had some problem with the presentation and my current understanding of the result might be partial. Yet, I have the impression that the result might have a (moderate) impact on the people working in causality.

**Q9 Complying With Reviewing Instructions:**

1: Yes.

---

### Decision · Program_Chairs · 2022-05-15

**Decision:**

Accept (Poster)

**Comment:**

Meta Review: This paper proposes tests, based on Verma constraints, of the front-door approach to causal effect identification. Testability is of particular interest for this approach because it is often criticized for making the seemingly unrealistic assumption that all mediators of the particular causal effect of interest are observed.

Pros

Reviewers agreed that this paper addresses an important problem and goes a long way towards practically relevant solutions to this problem. The paper was praised for its breadth and depth and for broadening its scope beyond the original front-door model. Authors also found the paper to be well written.

Cons

All reviewers initially felt that the presentation of the results was dense and at times difficult to follow, perhaps owing in part to the complexity of the material. However, the authors proposed specific modifications to their paper in response and three of the four reviewers responded they were satisfied with these suggestions. Overall there was a consensus to accept the paper.